# Structural basis for receptor binding and broader interspecies receptor recognition of currently circulating Omicron sub-variants

Zhennan Zhao[1,2,12], Yufeng Xie [1,3,12], Bin Bai[1,2,12], Chunliang Luo[1,4,12], Jingya Zhou[2,5,12], Weiwei Li[1,2,12], Yumin Meng[1,2], Linjie Li[1,2], Dedong Li[1], Xiaomei Li[6], Xiaoxiong Li[6], Xiaoyun Wang[1], Junqing Sun[1,4], Zepeng Xu[1,7], Yeping Sun [1], Wei Zhang[1], Zheng Fan[1], Xin Zhao [1], Linhuan Wu[8], Juncai Ma[8], Odel Y. Li[9], Guijun Shang[6], Yan Chai[1], Kefang Liu [1] ✉, Peiyi Wang [10] ✉, George F. Gao [1,2,5] ✉ & Jianxun Qi [1,2,11] ✉

Multiple SARS-CoV-2 Omicron sub-variants, such as BA.2, BA.2.12.1, BA.4, and BA.5, emerge one after another. BA.5 has become the dominant strain worldwide. Additionally, BA.2.75 is significantly increasing in some countries. Exploring their receptor binding and interspecies transmission risk is urgently needed. Herein, we examine the binding capacities of human and other 28 animal ACE2 orthologs covering nine orders towards S proteins of these sub-variants. The binding affinities between hACE2 and these sub-variants remain in the range as that of previous variants of concerns (VOCs) or interests (VOIs). Notably, R493Q reverse mutation enhances the bindings towards ACE2s from humans and many animals closely related to human life, suggesting an increased risk of cross-species transmission. Structures of S/hACE2 or RBD/ hACE2 complexes for these sub-variants and BA.2 S binding to ACE2 of mouse, rat or golden hamster are determined to reveal the molecular basis for receptor binding and broader interspecies recognition.

The outbreak of COVID-19 caused by severe acute respiratory syndrome coronavirus 2 (SARS-CoV-2) has posed a huge threat to public health and wreaked havoc on the world economy[1–3]. Since the SARS-CoV-2 Omicron variant emerged last year, it has rapidly swept the world and evolved into multiple sub-variants, including BA.1, BA.1.1, BA.2, BA.2.12.1, BA.3, BA.4 and BA.5[4–6]. Recently, BA.5 has overtaken previous strains and become the dominant strain. Meanwhile, BA.2.75 detected in many countries is increasing[7]. The trimeric spike (S)

protein is the major glycoprotein on the SARS-CoV-2 envelope and is responsible for engaging the ACE2 receptor for viral entry[8–10], and thus, it is a prime target for vaccine design and therapeutic antibody development[11–14]. Previous studies have shown that the S protein of BA.1 preferentially adopts a one-RBD-up conformation to engage hACE2[15–20], and its binding affinity to hACE2 is comparable to that of the SARS-CoV-2 prototype (PT) or slightly higher[15,17,21]. Compared to BA.1, the S protein of BA.2 adopts more open conformations (two-RBD-

[1]CAS Key Laboratory of Pathogen Microbiology and Immunology, Institute of Microbiology, Chinese Academy of Sciences, Beijing, China. [2]University of Chinese Academy of Sciences, Beijing, China. [3]Department of Basic Medical Sciences, School of Medicine, Tsinghua University, Beijing, China. [4]College of Veterinary Medicine, Shanxi Agricultural University, Jinzhong, China. [5]Research Network of Immunity and Health (RNIH), Beijing Institutes of Life Science, Chinese Academy of Sciences, Beijing, China. [6]Shanxi Academy of Advanced Research and Innovation, Taiyuan, China. [7]Faculty of Health Sciences, University of Macau, Macau, China. [8]Chinese National Microbiology Data Center (NMDC), Institute of Microbiology, Chinese Academy of Sciences, Beijing, China. [9]NHC Key Laboratory of Parasite and Vector Biology, National Institute of Parasitic Diseases, Chinese Center for Disease Control and Prevention, Shanghai, China. [10]Cryo-EM Center, Department of Biology, Southern University of Science and Technology, Shenzhen, China. [11]Beijing Life Science Academy, Beijing, China. [12]These authors contributed equally: Zhennan Zhao, Yufeng Xie, Bin Bai, Chunliang Luo, Jingya Zhou, Weiwei Li. ✉e-mail: liukf@im.ac.cn; wangpy@sustech.edu.cn; gaof@im.ac.cn; jxqi@im.ac.cn

up and three-RBD-up conformations) to bind hACE2 with a binding affinity that is two-fold higher than BA.1[22].

Recently, the apo S proteins from BA.2.12.1 and BA.4/5 were resolved by cryogenic electron microscopy (cryo-EM)[23], in which the S protein of BA.2.12.1 exhibits two conformational states corresponding to one-RBD-up and three-RBD-down (all-closed) conformations, whereas the S protein of BA.4/5 shows an all-closed conformation[23]. However, the structures of the BA.2.12.1 and BA.4/5 S protein complexed with hACE2 have not yet been solved.

Receptor-binding domain (RBD) in the S protein plays an essential role in ACE2 receptor recognition[10,24–26], and mutations in the RBD affecting receptor binding and viral infection have been reported[27]. For example, N501Y substitution observed in the RBDs of Alpha, Beta, Gamma and Omicron VOCs could enhance the receptor binding of ACE2s from human, mice and dog[28,29] but decrease the binding affinity between the SARS-CoV-2 RBD and ACE2s from the horse and intermediate host horseshoe bat[26,30]; G496S mutation in the BA.2 RBD leads to weakened hACE2 receptor binding[31]. Furthermore, the R346K substitution that occurred in the BA.1.1 RBD enhances the binding affinity with hACE2 by a mechanism of the "butterfly effect", indicating that mutations outside the binding interface could also have a major impact on receptor binding[31]. Compared to the RBDs of BA.1 and BA.2, there are several distinct substitutions in the BA.4/5 RBD, and the effect of these mutations on receptor binding remains to be studied.

The interspecies transmission of SARS-CoV-2 is an important issue that may accelerate the viral evolution and provide a source of new strain emergence[32]. For instance, Omicron BA.1 may be the product of the continuous evolution of SARS-CoV-2 in mice[33,34]. Twenty-four animal species have been reported to could be naturally infected by SARS-CoV-2, including cat, dog, mink, otter, ferret, lion, tiger, puma, snow leopard, gorilla, white-tailed deer, fishing cat, binturong, coatimundi, spotted hyena, Eurasian lynx, Canada lynx, hippo, hamster, mule deer, giant anteater, West Indian manatee, black-tailed marmoset, common squirrel monkey and big hairy armadillos (https://www.woah.org/en/document/86934/)[30,35,36]. With the occurrence of VOCs, the potential host spectra and cross-species transmission capability of SARS-CoV-2 might be dramatically changed. For example, substitutions of Q493K, Q498H and N501Y in RBD could promote the adaptation of SARS-CoV-2 in mice[37,38]; Y453F, F486L and N501T mutations in RBD were also observed in the mink-adapted strain[39]. Our recent work showed that Omicron BA.1 expanded its receptor binding spectra to the rodent, palm civet and various bat species compared with the PT and Delta variant[27]. Thus it is necessary to monitor the viral spillover in these species. As for BA.2, BA.2.12.1, and BA.4/5 carrying different substitutions compared with BA.1, they may have different interspecies receptor binding properties.

In this study, we measure the binding affinity between hACE2 and RBDs of BA.2, BA.2.12.1, BA.2.75, and BA.4/5 and explore the molecular mechanism of S or RBD of BA.2, BA.2.12.1, BA.2.75, or BA.4/5 bound to hACE2. We then evaluate the receptor binding capability of the BA.2 and BA.4/5 RBDs to other 28 animal ACE2 orthologs. We found that the binding affinities of the SARS-CoV-2 PT and its variants towards hACE2 fall into a narrow nanomolar range. The reverse mutation of R493Q in RBD of BA.4/5 could change the surface electrostatic of the receptor-binding motif (RBM), thereby enhancing its binding affinities to ACE2s of human, rabbit, horse, pig, goat and sheep. Therefore, BA.4/5 may have the potential to increase interspecies transmission. Furthermore, the structural bases for receptor binding and broader interspecies receptor recognition are provided to understand interspecies transmission at the atomic level.

## Results
### The receptor binding and Pseudovirus entry of BA.2, BA.2.12.1, BA.2.75, and BA.4/5
Compared with the SARS-CoV-2 PT, there are 31, 33, 37, and 34 amino acid (AA) mutations in the S protein of BA.2, BA.2.12.1, BA.2.75, and

BA.4/5, respectively (Supplementary Fig. 1). Notably, the S protein of some early strains of BA.4/5 harbors N658S substitution (Supplementary Fig. 1)[23]. Compared with BA.2, BA.2.12.1 has extra L452Q and S704L substitutions in the S protein, while BA.4/5 has extra △69H-70V, L452R and F486V mutations (Supplementary Fig. 1). As for BA.2.75, there are more distinct substitutions, including K147E, W152R, F157L, I210V, G257S, G339H, G446S, and N460K (Supplementary Fig. 1). Moreover, the residue 493 in the BA.4/5 and BA.2.75 S proteins was reversely mutated to Q from R (Supplementary Fig. 1).

Since the appearance of the Omicron variant, several Omicron sub-variants emerged, among which Omicron BA.1 was the predominant pandemic strain at the beginning, and the following BA.1.1 was soon replaced by BA.2 (Fig. 1a). Sequentially, BA.2.12.1 emerged and expanded substantially (Fig. 1a) and BA.5 became the predominant pandemic strain recently (Fig. 1a).

The binding affinities between hACE2 and RBDs from the five Omicron sub-variants (BA.2, BA.2.12.1, BA.2.75, and BA.4 and BA.5) were analyzed by a surface plasmon resonance (SPR) assay (Supplementary Table 1). The affinity of the PT RBD to hACE2 was $23.8 \pm 2.6$ nM (Fig. 1b), consistent with previous reports[10,21,29]. The $K_D$ of BA.4/5 RBD binding to hACE2 was approximately $9.0 \pm 1.7$ nM, which is ~2.6- and ~1.6-fold higher than that of the PT ($23.8 \pm 2.6$ nM) and BA.2 ($14.6 \pm 2.1$ nM) RBDs, respectively (Fig. 1b), while BA.2.75 RBD ($7.5 \pm 0.2$ nM) has a similar binding capacity for hACE2 to BA.4/5 ($9.0 \pm 1.7$ nM) (Fig. 1b). However, the binding affinity of BA.2.12.1 ($27.4 \pm 0.9$ nM) to hACE2 was ~1.9-fold lower than that of BA.2 ($14.6 \pm 2.1$ nM) (Fig. 1b). Interestingly, although various SARS-CoV-2 variants harbor distinct mutations in their RBD, the binding affinities between their RBDs and hACE2 are in a narrow range from one digit to two digits of nanomolar, which is comparable to the binding affinity between PT and hACE2 (Fig. 1c). Next, we carried out a viral entry assay of Vesicular stomatitis viruses (VSV) pseudotyped by spike proteins of SARS-CoV-2 Omicron sub-variants. It showed that these pseudoviruses could enter the Vero cells with different capabilities, among which the BA.2.75 pseudovirus has the highest infection efficiency, and BA.4/5 follows, whereas BA.3 pseudovirus has the lowest infection efficiency (Fig. 1d).

### The Structures of BA.2, BA.2.12.1, BA.2.75 and BA.4/5 (N658S) S or RBD in Complex with hACE2
To unravel the underlying molecular mechanism of the S proteins of Omicron BA.2, BA.2.12.1, BA.2.75, and BA.4/5 bound to hACE2, we determined their atomic structures of the S (BA.2, BA.2.12.1 or BA.4/5) or RBD (BA.2.75) in complex with hACE2 by cryo-EM or X-ray crystallography (Supplementary Figs. 2–4 and Supplementary Tables 2 and 3). In complex structures, all S proteins of BA.2, BA.2.12.1 and BA.4/5 (N658S) exhibit a three-RBD-up conformation (Supplementary Figs. 2-4). To understand the detailed interactions between hACE2 and these RBDs, focus refinement on RBD/hACE2 was performed, and local maps of BA.2 RBD/hACE2, BA.2.12.1 RBD/hACE2 and BA.4/5 RBD/hACE2 were resolved at resolutions of 3.14 Å (Supplementary Figs. 2 and 5), 3.09 Å (Supplementary Figs. 3 and 5) and 2.66 Å (Supplementary Figs. 4, 5), respectively. The crystal structure of BA.2.75 RBD in complex with hACE2 was determined at a resolution of 2.9 Å (Supplementary Fig. 5).

According to our previous reports[10,40], the binding interface between RBD and ACE2 could be divided into two patches (patch 1 and patch 2). As for patch 1, these four omicron sub-variants show different interaction networks (Fig. 2a). Residue N477 of BA.2, BA.2.75 and BA.4/5 RBD forms an H-bond with S19 of hACE2, while A475 engages S19 with an H-bond in the BA.2.12.1 RBD/hACE2 and BA.2.75/hACE2 complexes. N487 in the RBD of BA.2 and BA.4/5 interacts with both Q24 and Y83 of hACE2 through H-bonds, but it only forms an H-bond with Y83, and its interaction with Q24 was absent in complex structures of hACE2 and RBD of BA.2.12.1 and BA.2.75. Interestingly, the side chain of H34 adopts two alternative conformations in the BA.4/5 RBD/hACE2 complex, in contrast to the

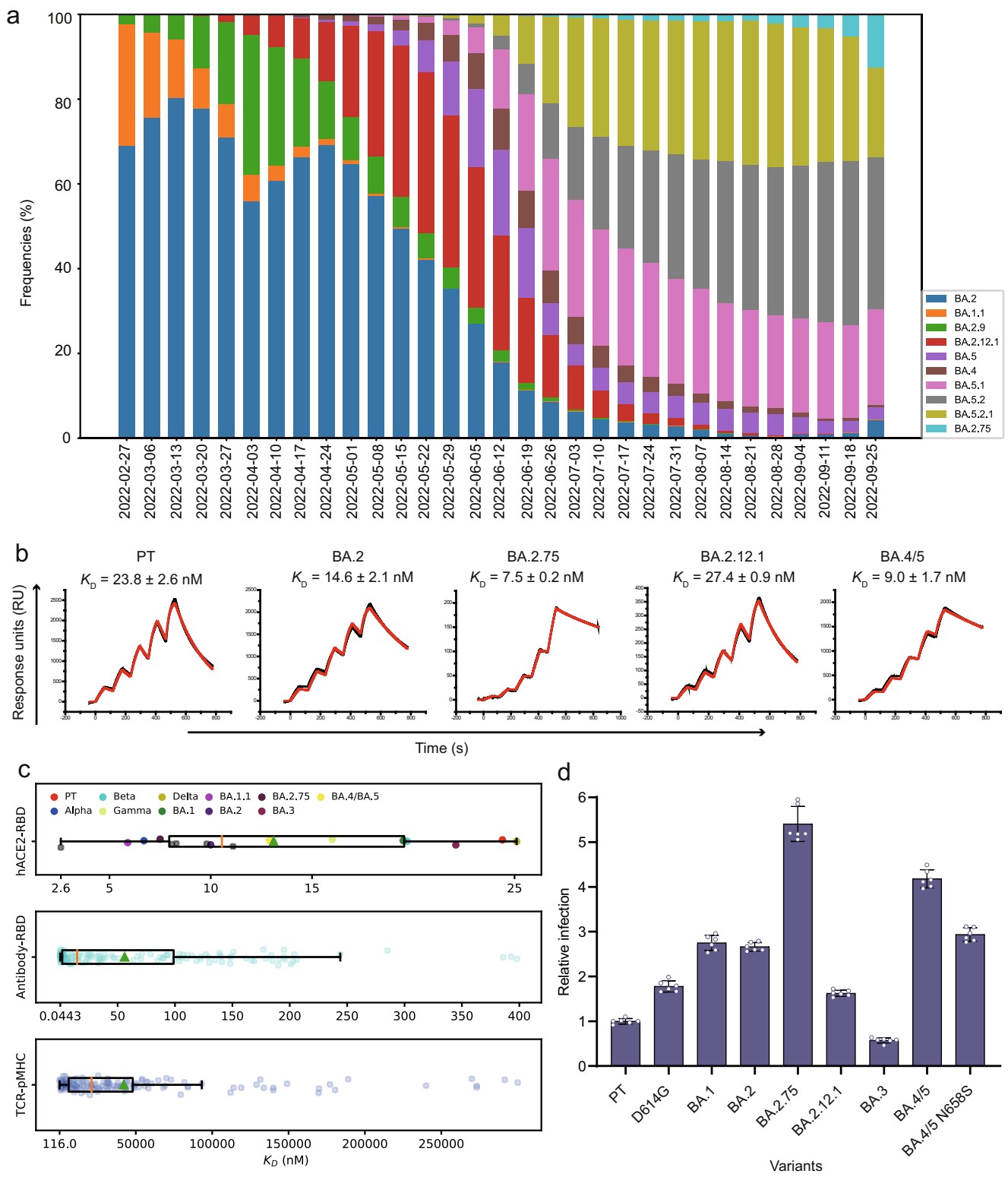

other three complexes. H34 of hACE2 forms an H-bond with Y453 of the BA.4/5 RBD or S494 of the BA.2.12.1 RBD. Notably, R493 of BA.2 and BA.2.12.1 forms a salt bridge with E35 from ACE2. While in BA.2.75 and BA.4/5 RBD, R493 is substituted by Q493, which is hydrogen-bonded with K31 in the BA.4/5 RBD/hACE2 complex. Although the H-bonds were absent between Q493 and K31 in the BA.2.75 RBD/hACE2 complex, there are more Van der Waals' contacts between Q493 of BA.2.75 and K31 of hACE2 than that between R493 of BA.2 and K31 of hACE2 (Supplementary Table 4). In addition, substitution

F486V in the BA.4/5 RBD decreases its hydrophobic interactions with L79, M82, and Y83 in hACE2.

Contrary to the altered interactions in patch1, the interaction network in patch 2 of these three RBDs is almost the same (Fig. 2b). Consistently, residues Y449, R498, T500, and G502 in these four RBDs form hydrophilic interactions with D38, Y41 and K353. However, there are still subtle differences in the interaction network in patch 2. Q42 of hACE2 forms an additional H-bond with R498 of BA.2 RBD. T500 of hACE2 is hydrogen-bonded with D355 in the RBDs from BA.2, BA.2.12.1

**Fig. 1 | Prevalence and receptor binding characteristics of SARS-CoV-2 Omicron sub-variants. a** Frequencies of BA.2 BA.2.12.1, BA.2.75, BA.4, BA.5 and other five Omicron sub-variants deposited in the Global Initiative on Sharing All Influenza Data (GISAID) by time as indicated. **b** The SPR curves for the prototype (PT), BA.2, BA.2.75, BA.2.12.1, and BA.4/5 RBDs binding to hACE2. Raw and fitted curves are represented by black and red lines, respectively. $K_D$ values shown are the mean ± standard deviations (SD) of three independent experiments. **c** The range of binding affinities between hACE2 and RBD, antibody and RBD, or TCR and pMHC. The binding affinities between hACE2 and RBD from the PT or different SARS-CoV-2 variants, collected from previously reported results[21,28,31] and this study, were shown as circles with indicated colors (*n* = 16). The binding affinity data between antibody and RBD (*n* = 180), or TCR and pMHC (*n* = 249), which were respectively derived from COVIC-DB (https://covicdb.lji.org/) and ATLAS (http://atlas.wenglab.org/web/index.php), were also statistically analyzed, and each circle represented one antibody-RBD or TCR-pMHC binding affinity value. Perpendicular orange lines

and green triangles present medians and means, respectively. The boundaries of the box in each boxplot are the 25th (Q1) and 75th (Q3) percentiles of the dataset; the minima of each box plot is the minimum in the dataset that is larger than or equal to Q1 − 1.5 * the interquartile range (IQR), whose value is equal to Q3 - Q1; the maxima is the maximum in the dataset that is less than or equal to Q3 + 1.5 * IQR; the lower whisker is the difference between Q1 and the minimum, and the upper whisker is the difference between the maximum and Q3. **d** Pseudovirus entry assay for the PT, D614G and Omicron sub-variants. Pseudoviruses for the PT and variants were respectively diluted to an equal amount, and each pseudovirus was added to the wells (*n* = 6) containing Vero cells. After the 15 h incubation, the fluorescent cells for each well were counted using a CQ1 confocal image cytometer (Yokogawa). Infectivity for D614G and each Omicron sub-variant was normalized at the basis of the PT, and the relative infection fold was shown as the y-axis. Data were presented as mean values ± SD. Two independent experiments were performed with similar results. Source data are provided as a Source Data file.

and BA.4/5. An additional H-bond between T500 of hACE2 and R357 of the BA.2 or BA.2.12.1 RBD was also observed. Compared with the residue N501 in PT RBD, Y501 in these four Omicron RBDs forms π-π stacking interaction with Y41 in hACE2, contributing to the increased binding affinity towards hACE2.

In our previous study, T478K, Q493R, Q498R, and E484A substitutions on the receptor binding interface of Omicron BA.1 RBD could change electrostatic charges to strongly positive, affecting the binding between RBD and hACE2[21]. However, R493Q reverse mutation was observed in BA.2.75 and BA.4/5 sub-variants, which could weaken the positive charge on the binding interface and may affect the binding to some extent (Fig. 2c).

## Mutagenesis analysis of the key residues for hACE2 binding in the RBDs of BA.2, BA.2.12.1, and BA.4/5

Compared with BA.2 RBD, BA.2.12.1 RBD only has one unique mutation site at residue 452 (L452Q), which is far from the receptor binding interface and does not interact with hACE2 directly. However, the binding affinity between BA.2.12.1 RBD and hACE2 is slightly lower than that of BA.2 RBD (Fig. 1b), and structural analysis also showed some differences between BA.2 RBD and BA.2.12.1 RBD in complex with hACE2. For instance, N477 in the BA.2 RBD forms H-bonds with hACE2, which is absent in the BA.2.12.1/hACE2 complex (Fig. 3a and Supplementary Table 4). In contrast, A475 and S494 in the BA.2.12.1 RBD contact hACE2 through H-bonds, which do not exist in the BA.2/hACE2 complex (Fig. 3a and Supplementary Table 4). In addition, the number of H-bonds formed by N487, Y449 and R498 from the BA.2 RBD in the complex is more than that of the BA.2.12.1 RBD/hACE complex (Fig. 3a and Supplementary Table 4).

Structural comparison between BA.2.75 RBD and BA.2 RBD in individual complex showed that A475 of BA.2.75 RBD also forms polar contacts with hACE2, which was not observed in the BA.2/hACE2 complex (Fig. 3b and Supplementary Table 4). However, the H-bond contributed by R493 of the BA.2 RBD was absent and was replaced by Van der Waals' contacts in the BA.2.75 RBD/hACE2 complex (Fig. 3b and Supplementary Table 4).

Compared with RBD from BA.2, the RBD of BA.4/5 has three distinct mutations (L452R, F486V and R493Q) (Supplementary Fig. 1), which displays different binding patterns in the complex interface (Fig. 3c). To evaluate the effect of these substitutions on receptor binding, we mutated these three residues from the BA.4/5 RBD to corresponding residues in BA.2 RBD one by one and performed the binding assay (Fig. 3d and Supplementary Table 5). The binding affinity between BA.4/5 RBD harboring R452L mutation and hACE2 was about ~14 nM, similar to that between wild-type BA.4/5 RBD and hACE2 (12.9 ± 1.8 nM) (Fig. 3d). After performing mutagenesis analysis of the V486F for the BA.4/5 RBD, there was a 3.2-fold increase in its binding affinity towards hACE2 (Fig. 3d). The structural analysis shows residue F486 forms hydrophobic contacts with L79, M82 and Y83 of the hACE2

receptor in the BA.2 RBD/hACE2 complex (Fig. 3e), whereas V486 of the BA.4/5 RBD has a smaller sidechain than F486, resulting in decreased hydrophobic interactions (Fig. 3e). The binding assay showed that the binding affinity for hACE2 decreased ~9.8 fold when Q493 of BA.4/5 was mutated to R493 (Fig. 3d). In the BA.4/5 RBD/hACE2 complex, Q493 forms an H-bond with hACE2, as observed in other variants carrying this mutation (Fig. 3f). Furthermore, Y453, which is close to Q493, forms an additional H-bond with hACE2 in BA.4/5 RBD/hACE2 complex (Fig. 3f). In the BA.2 RBD/hACE2 complex, both R493 of RBD and K31 of hACE2 are positively charged, which could decrease their binding, although a salt bridge was observed between R493 of RBD and E35 of hACE2 (Fig. 3f). Altogether, R493Q substitution could improve the binding capacity towards hACE2.

## The receptor binding spectra of omicron BA.2 and BA.4/5

Our recent work suggested Omicron BA.1 has a broader-species receptor binding[27]. To explore whether the host range of BA.2 and BA.4/5 was changed, we performed flow cytometry assay to evaluate their receptor binding capacities, in which 29 ACE2 orthologs covering Primates (human, monkey, grivet, chimpanzee and gorilla), Lagomorpha (rabbit), Rodentia (mouse, rat, guinea pig and golden hamster), Pholidota (Malayan pangolin), Perissodactyla (horse), Carnivora (cat, dog, fox, civet, and mink), Artiodactyla (goat, sheep, pig, alpaca, bovine and camel), Chiroptera (little brown bat, fulvous fruit bat, greater horseshoe bat, Chinese horseshoe bat and least horseshoe bat) and Afrotheria (lesser hedgehog tenrec) were measured (Supplementary Fig. 6).

Flow cytometry assay showed that BA.2 RBD has a similar receptor binding spectra to BA.1. Notably, the cells transfected with full-length ACE2 from the rabbit, rat, golden hamster, cat, horse, pig, goat and sheep have a higher binding capacity for BA.4/5 RBD than that of BA.1 or BA.2 (Fig. 4). Given that the flow cytometry assay is semi-quantitative and could not precisely reflect the binding affinity, we further conducted an SPR assay to measure the binding affinities of these ACE2 orthologs (mouse and dog ACE2 included as well) towards BA.1, BA.2 and BA.4/5 RBDs. (Supplementary Table 6). Compared to BA.2 RBD, the binding affinities of BA.4/5 RBD to ACE2 from rabbit, horse or pig and ACE2 from sheep or goat increased more than 10-fold and 3-fold, respectively (Fig. 5a–c). BA.2 RBD and BA.4/5 RBD show similar binding affinity to rat, golden hamster, cat and dog ACE2s (Fig. 5a–c). Only mouse ACE2 (mACE2) decreased its binding affinity to BA.4/5 RBD.

Given that the R493Q mutation increases the binding capacity of BA.4/5 or BA.2.75 RBD to hACE2, the effect of this substitution on interspecies transmission was also evaluated. The binding affinities of these ACE2 orthologs to BA.2 RBD harboring the R493Q reverse mutation showed enhanced binding affinities to ACE2s of rabbit, horse, pig, goat, and sheep but displayed significantly decreased binding capacities for ACE2s of mouse, golden hamster and dog (Fig. 5a–c).

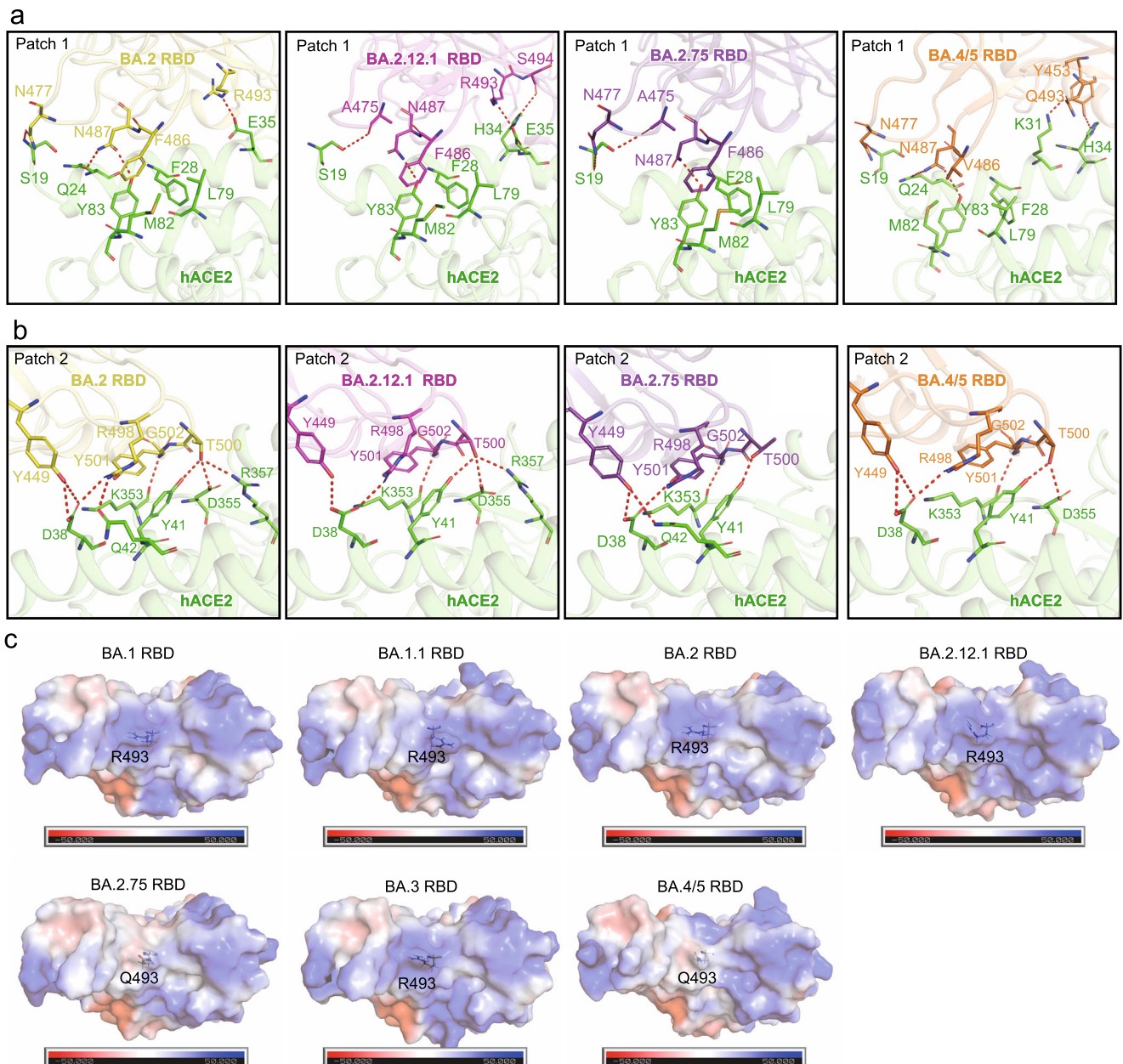

**Fig. 2 | The interaction analysis of BA2, BA.2.12.1, BA.2.75 and BA.4/5 RBDs bound to hACE2. a**, **b** Polar interactions of BA2 (yellow), BA.2.12.1(hot pink), BA.2.75 (purple) and BA.4/5 RBDs (orange) with hACE2 (green), which were analyzed at a cutoff of 3.5 Å. Key residues were shown as sticks. **c** Electrostatic surface of Omicron BA.1, BA.1.1, BA.2, BA.2.12.1, BA.2.75, BA.3, and BA.4/5 RBDs. Residue R493 or Q493 in these RBDs was shown as sticks.

## Structures of the omicron BA.2 RBD in complex with mACE2, RatACE2, or ghACE2

The house mice, pet hamsters and rats live closely with human, which poses potential risks of interspecies transmission of SARS-CoV-2 and its variants, and thus mouse/rats/hamsters-to-human transmission routes deserve to be continuously monitored. Previous studies showed that the SARS-CoV-2 PT could not infect mice, whereas omicron BA.1 can achieve close-contact transmission in mice and result in severe lung lesions and inflammatory responses, suggesting Omicron has the potential for transmission from mouse to human[41]. As for hamster, it is susceptible to SARS-CoV-2 infection[42,43]. It has been reported that hamsters could be naturally infected by Delta VOC as well[44,45]. However, in our study, mice, rats and golden hamsters displayed an opposite response (decreased binding affinity) to the R493Q reverse mutation compared with other species (human, rabbit, horse, pig, goat and sheep). In addition, as rodents, they showed significantly

different binding capacities for Omicron variants (Fig. 5a–c). To explore the reason causing these differences, we determined the complex structures of BA.2 S in complex with mouse, rat and golden hamster ACE2s (Supplementary Figs. 7–9). As described above, the binding interface of the RBD/ACE2 complex could be divided into two patches. In patch 1, residue N487 of the BA.2 RBD forms H-bonds respectively with Q24 and Y83 of ghACE2, which was not observed in the BA.2 RBD/mACE2 and BA.2 RBD/RatACE2 complex (Fig. 6a–c). Furthermore, S494 of BA.2 RBD forms hydrogen-bonded with Q34 of ghACE2, which is replaced by the H-bond between R493 of the RBD and Q34 of the ACE2 in the BA.2 RBD/RatACE2 complex (Fig. 6a, c). In the BA.2 RBD/mACE2 complex, two H-bonds with N31 of mACE2 were contributed by R493 (Fig. 6b). In these three complexes, F486 of the BA.2 RBD forms hydrophobic contacts with Y83 of the ACE2. Notably, hydrophobic contact between residue L79 of the ghCE2 and I79 of the RatACE2 may also stabilize the interface (Fig. 6a–c). In patch 2,

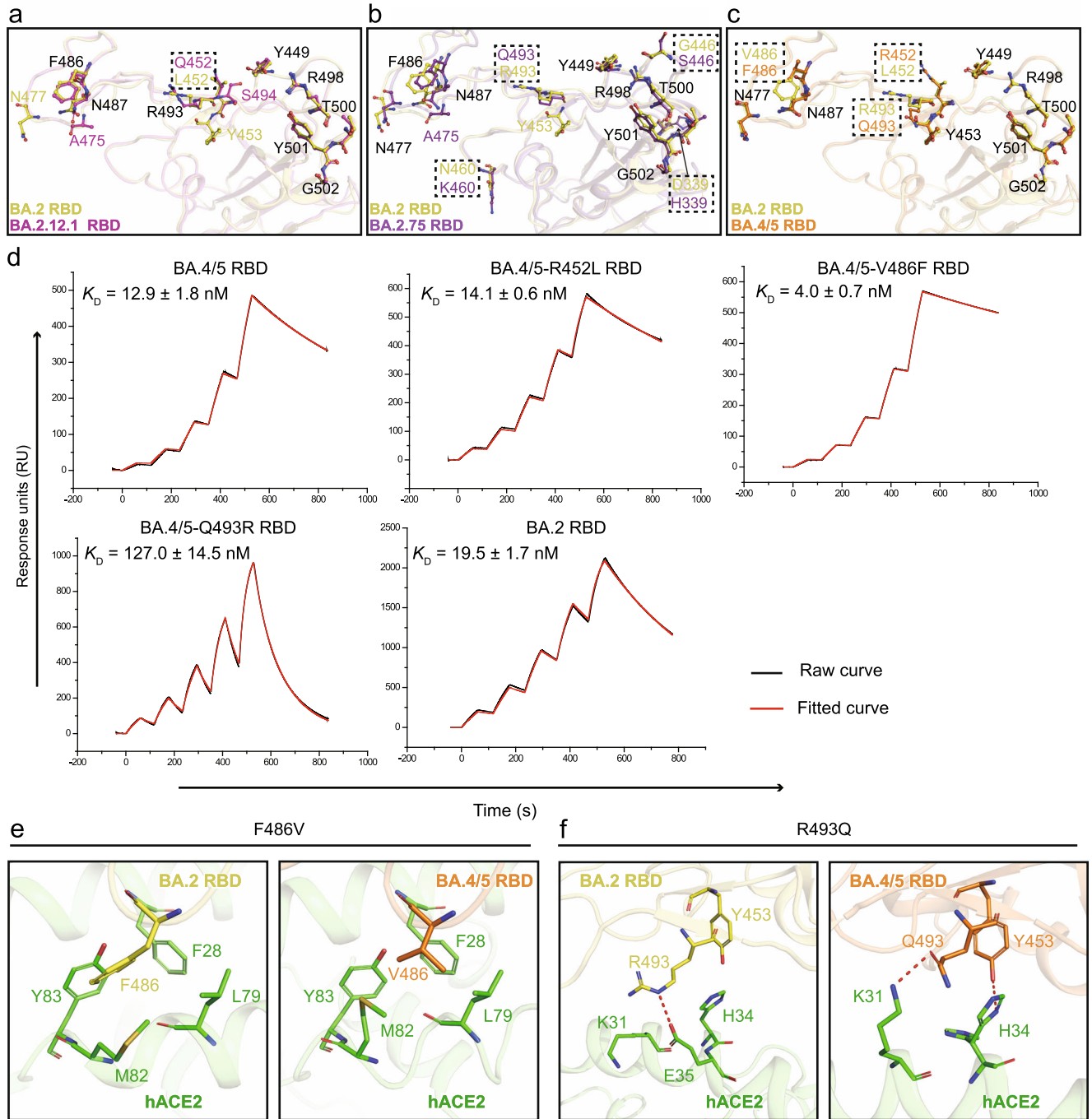

**Fig. 3 | Key residues for binding the hACE2 receptor. a–c** Structural comparisons between Omicron BA.2 and BA.2.12.1 RBDs, BA.2 and BA.2.75 RBDs, or BA.2 and BA.4/5 RBDs. BA.2, BA.2.12.1, BA.2.75, and BA.4/5 RBDs were colored corresponding to Fig. 2a, b. Differential residues between the two variants for comparison were boxed in the dashed lines and shown as sticks in their respective colors. Key residues were also presented as sticks, and those shared by two variants for comparison were labeled in black, except for differential residues, otherwise were labeled with their respective colors. **d** The binding affinities of the BA.2, BA.4/5 and three BA.4/5 RBD mutants harboring R452L, V486F or R493Q towards the hACE2 receptor. Raw and fitted curves are represented by black and red lines, respectively. $K_D$ values shown are the mean ± SD of four independent experiments. **e, f** The impact of substitutions F486V and R493Q on the interactions between RBD and hACE2. hACE2, BA.2 and BA.4/5 RBDs were colored corresponding to Fig. 2a, b. Source data are provided as a Source Data file.

residues Y449, R498, T500, Y501, and G502 of BA.2 RBD form an extensive H-bond network with D38, Y41 and K353 of the ghACE2 (Fig. 6a). But only T500 and G502 of BA.2 RBD form two H-bonds with Y41 and H353 in the BA.2 RBD/mACE2 complex (Fig. 6b). In the BA.2 RBD/RatACE2 complex, an extra H-bond is formed between Y501 of the BA.2 RBD and H353 of the RatACE2 compared with the BA.2 BRD/mACE2 (Fig. 6c). Therefore, it seems that more polar interactions are occurring in ghACE2/BA.2 RBD interface, consistent with the SPR result

that ghACE2 has a higher binding affinity for BA.2 RBD than that of ratACE2 and mACE2.

Compared with hACE2, H34 is replaced by Q34 in ghACE2, mACE2 and RatACE2 (Fig. 6d). Q34 of ghACE2 forms an H-bond with S494 of the BA.2 RBD (Fig. 6e), and R493 of BA.2 RBD forms Van der Waals' contacts with Q34 and K31 of ghACE2 (Supplementary Table 7). SPR results showed that R493Q mutation decreased the binding affinity between BA.2 RBD and ghACE2 by ~8.2-fold (Fig. 5a–c). Given that

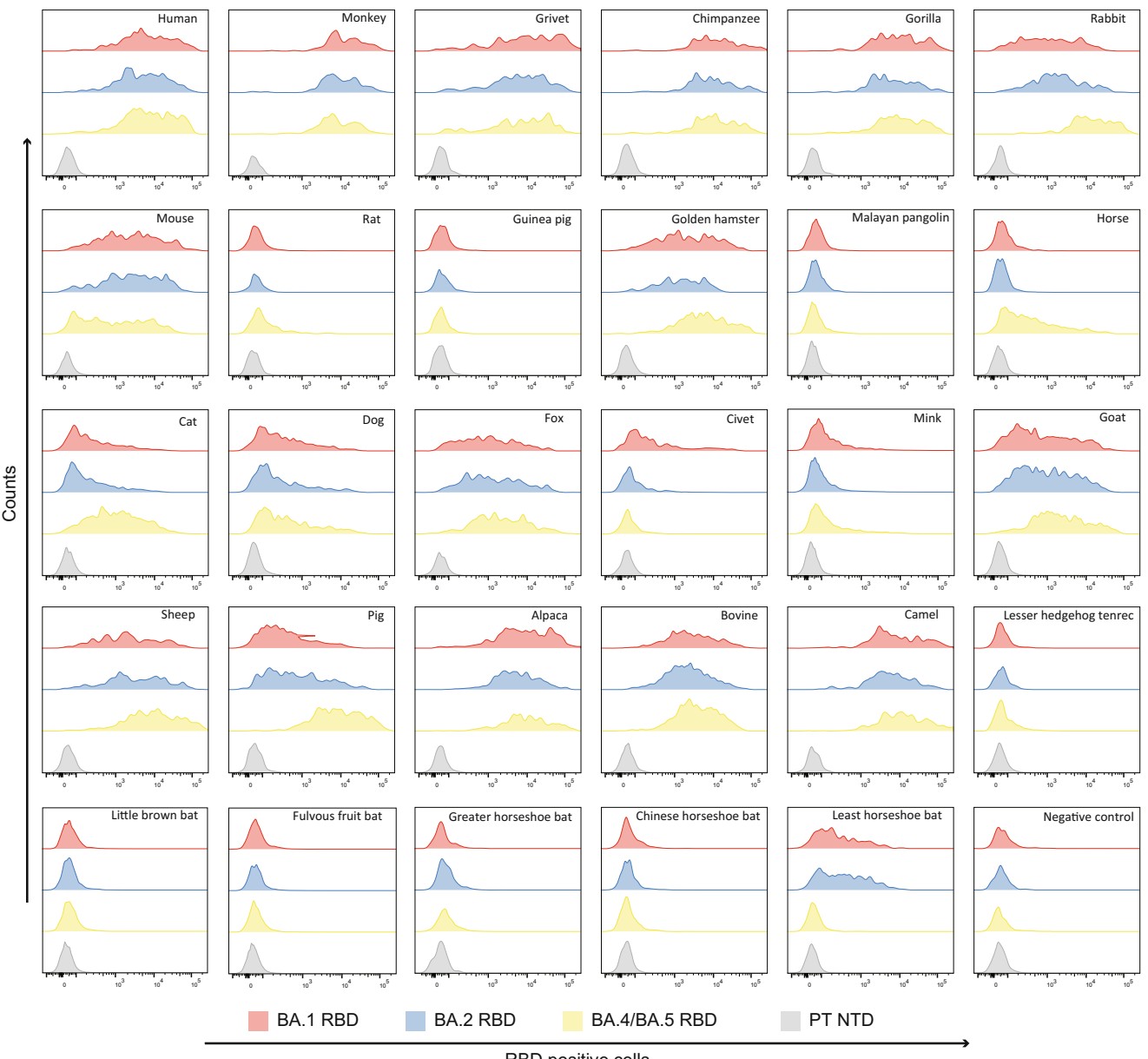

**Fig. 4 | The binding between 29 ACE2 orthologs and RBDs from BA.1, BA.2 and BA.4/5 detected by flow cytometry.** BHK-21 cells expressing eGFP-fused full-length ACE2s from 29 species, including humans, were incubated with BA.1, BA.2, and BA.4/5 RBDs with a His-tag in their C-terminus, respectively. SARS-CoV-2 PT NTD with a His-tag was used as the negative control for RBDs of BA.1, BA.2, and BA.4/5, and eGFP-fused full-length human CD26 was used as the negative control for ACE2s. The anti-His tag mouse monoclonal antibody conjugated to APC was used to detect His-tagged proteins. Three independent experiments were performed with similar results.

Q493 has a relatively shorter side chain than R493, R493Q substitution may weaken interactions between R493 of BA.2 RBD and Q34 (and K31) of ghACE2. In the BA.2 RBD/RatACE2 complex, R493 of RBD forms an H-bond with Q34 (Fig. 6e). The binding affinity decreased ~2-fold with the Q493 substitution at this site (Fig. 5a–c). K31 is a very conserved residue among hACE2, ghACE2 and RatACE2, but it was substituted by N31 in mACE2 (Fig. 6d) and formed two H-bonds with BA.2 RBD (Fig. 6e). When R493 of BA.2 was substituted by Q493, these H-bonds were gone, resulting in a decrease in their binding affinity (Fig. 5a–c).

## Discussion
Previous work shows that the apo S of BA.2 and BA.2.12.1 exhibits two conformations: close conformation with all three RBDs down and partial open form with one RBD up[23]. As for BA.4/5 (N658S) S-trimers, it adopts a close or semi-close form[23]. RBDs of BA.2, BA.2.12.1, and BA.4/5

(N658S) can be induced to stand up to interact with hACE2, as reported in the MERS-CoV and SARS-CoV S proteins[46]. The cryo-EM structure of BA.2 S in complex with hACE2 has been reported that two or three RBDs adopted "up" conformation to contact with hACE2[22]. The BA.2 S/mACE2 complex was also determined, in which the BA.2 S protein adopted one- or two-RBD-up conformation[22]. In our results, three-RBD-up conformation was observed to bind the hACE2 or mACE2 receptor. These conformational differences may be due to the differences in S-protein constructs. In addition, the contact network of BA.2 RBD with hACE2 slightly differs from that determined by another group[22]; and fewer interactions were observed in our BA.2 RBD/mACE2 complex compared to the published data[22], both of which may be caused by the intrinsic flexibility of the protein in aqueous. The BA.2 RBD/hACE2 structures solved by crystallography and cryo-EM also have some differences. Our previous study reported the crystal structure of

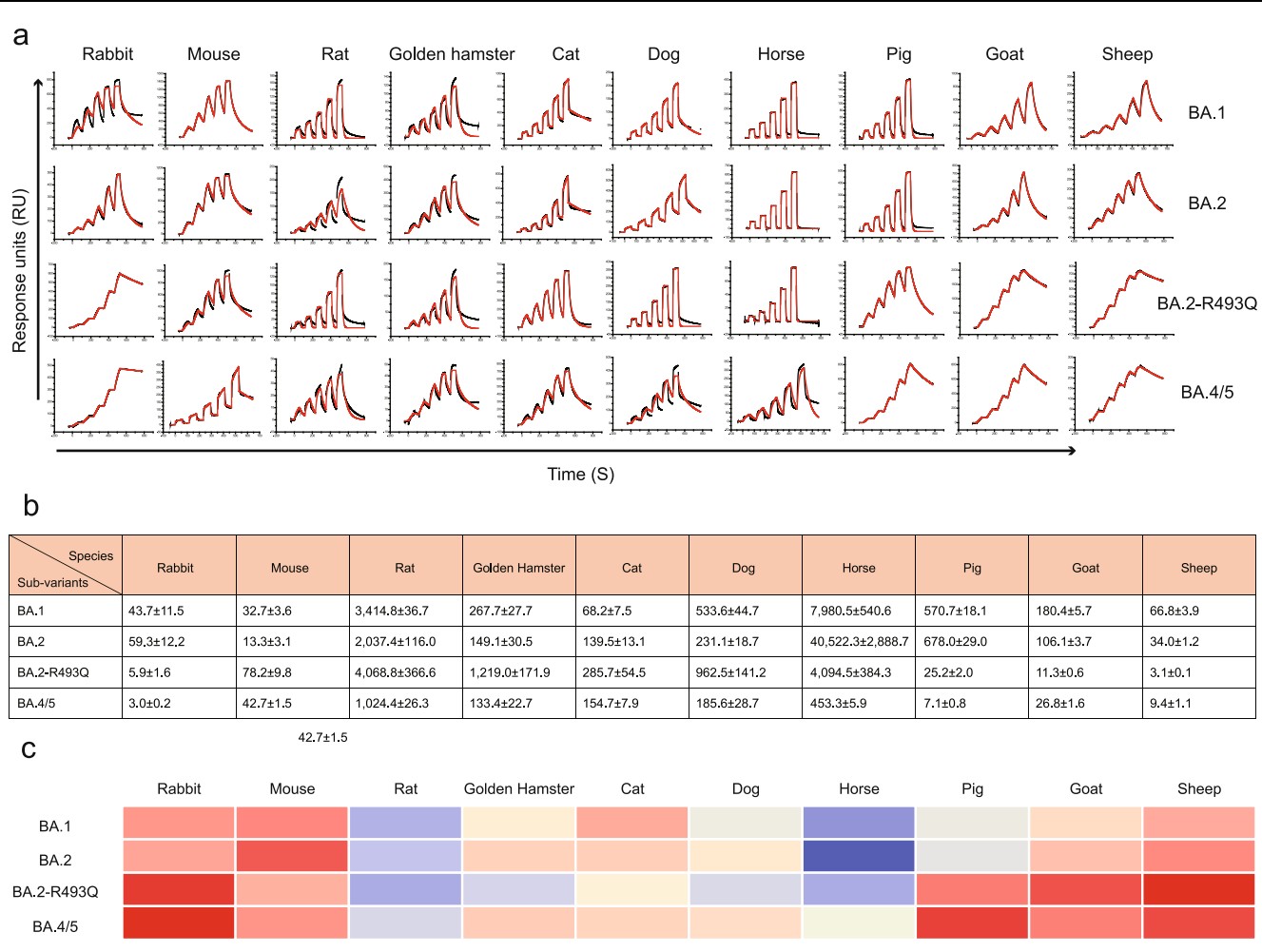

**Fig. 5 | The binding affinities between ACE2 orthologs and BA.1, BA.2, a BA.2 mutant harboring the R493Q substitution (BA.2-R493Q) or BA.4/5 RBD. a** The SPR curves for the BA.1, BA.2, BA.2-R493Q, and BA.4/5 RBDs binding to ACE2 ortholog in ten species were shown. Raw and fitted curves are represented by red and black lines, respectively. **b** The statistical table of binding affinities ($K_D$, nM). **c** A heatmap was used to present the binding affinities. The logarithm of each value (nM) for binding affinity corresponds to the indicated colors. SPR assay was repeated three times with similar results. Source data are provided as a Source Data file.

BA.2 bound to hACE2 and in the crystal structure[31], N90-glycan of hACE2 could form an H-bond with T415 in the BA.2 RBD, which, however, was not observed in the cryo-EM structure of BA.2 S/hACE2 determined by us here or another group[22].

In our previous work, we speculated that R493 and R498 of RBD might be attracted by the negative charges around E35 and D38 of hACE2, respectively, based on BA.1 RBD/hACE2 structural analyses[21]. A study from another group considered that Q493R substitution could increase the binding affinity of BA.2 RBD to hACE2 based on their structural information[22]. The SPR measurements in our later work suggested that Q493R single-point mutation in the PT RBD decreases the binding affinity to hACE2[27], consistent with the results obtained by this paper and another group[47]. The cryo-EM structure of BA.4/5 RBD in complex with hACE2 in this paper further indicates that R493Q reverse mutation makes K31 and its near residues of hACE2 more approach the RBD and form more Van der Waals' contacts and H-bonds, contributing to the enhanced binding affinity between RBD and hACE2.

Furthermore, the SPR assay shows that the BA.2-R493Q mutant increases the binding affinities of RBD to several animal ACE2s, including rabbit, horse, pig, goat and sheep. BA.4/5 harboring the R493Q substitution has an equivalent or even higher binding capacity for these species than BA.2-R493Q, suggesting BA.4/5 sub-variant may have increased interspecies transmission risk from natural hosts to

these domestic animals than BA.2. Notably, BA.2-R493Q mutant has a significantly decreased binding affinity towards mACE2 or golden hamster, and slightly impairs the binding capacities of BA.2 to ratACE2. The previous study demonstrated that Q493H or Q493K substitution was observed in the mouse-adaptive SARS-CoV-2 strains[37,48]. These findings imply that R493Q reverse substitution may not be an adaptive mutation for rodents.

Currently, 25 species have been reported to be naturally infected by SARS-CoV-2[30,35,36]. Furthermore, SARS-CoV-2 could broadly recognize numerous ACE2 orthologs[25]. Our previous work indicated that SARS-CoV-2 RBD could bind to ACE2s from marine mammals[48], which have similar contact interfaces with hACE2. The risk of marine mammals infected by BA.4/5 may be further increased, which deserves to monitor continuously.

The L452R mutation among VOCs was first reported in the Delta VOC. Structure analysis indicates that L452R is far from the receptor binding motif (RBM). The binding affinity between PT RBD L452R substitution and hACE2 varies slightly[28]. However, Residue 452 mutation was never observed in the early dominated Omicron sub-variants such as BA.1, BA.1.1, BA.2 and BA.3, but it emerged in the most recent circulating strains like BA.2.12.1, BA.4 and BA.5. In this study, we found the binding affinity to hACE2 was also changed no more than two folds regardless of substitution by Q452 (BA.2.12.1 RBD) or R452 (BA.4/5

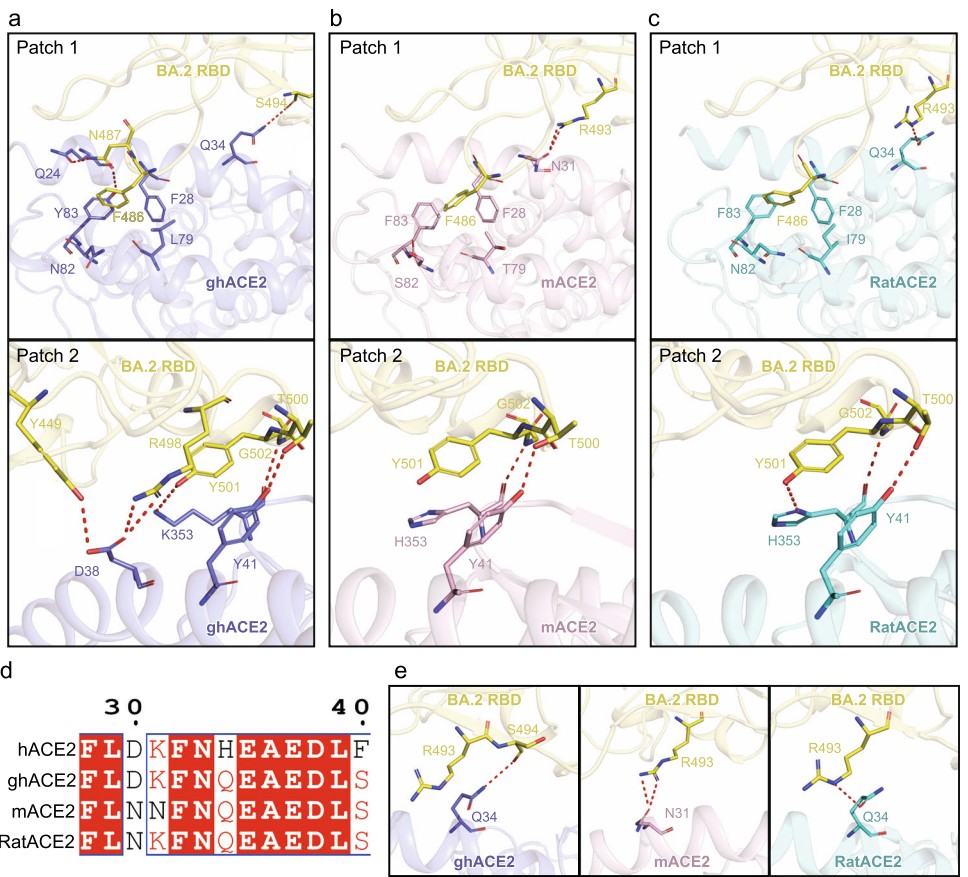

**Fig. 6 | BA.2 RBD in complex with ghACE2, mACE2 or RatACE2. a–c** The polar interactions between BA2 RBD (yellow) and ghACE2 (medium slate blue) **a**, mACE2 (pink) **b**, or RatACE2 (cyan) **c**. Key residues were shown as sticks. The polar interactions were analyzed at a cutoff of 3.5 Å. **d** Sequence alignment of hACE2, ghACE2, mACE2 and RatACE2. The alignment was performed by T-COFFEE and visualized by ESPript 3. **e** Structural comparison of residue R493 of BA.2 contact with ghACE2, mACE2 and RatACE2.

RBD). L452 mutation is also considered an immune escape hotspot[49]. Previous work indicates that the L452 mutation escapes some monoclonal antibodies or the antibodies in the sera of convalescent or vaccinated people[49]. Furthermore, mutation at this site also affects the cellular immune response against SARS-CoV-2 by resistance to HLA-restricted antigen recognition[50,51].

Although the RBDs of SARS-CoV-2 VOCs have various degrees of substitutions, the binding affinity between various RBDs and hACE2 only changes slightly, ranging from nM to tens of nM, which is optimal for SARS-CoV-2 virus entry. It seems that for the ligand-receptor binding, the affinities always fall into a reasonable range, similarly to the fact that the affinity between major histocompatibility complex class I (MHC I) and T cell receptors are always in one or two digits µM[52–55]. K417N, G446S and E484A mutations of BA.1 RBD decrease the binding affinity of RBD to hACE2 but play pivotal roles in the immune escape[27,56,57]. To keep the sufficient binding affinity, R493Q reverse substitution and N501Y mutation in the RBD increase the binding affinity to hACE2[27]. Thus, mutations that emerged in the SARS-CoV-2 RBD are the balance between immune escape and receptor binding.

In this study, the VSV-based pseudovirus entry assay was performed according to the previous protocols[58]. The pseudoviruses harboring S protein from different SARS-CoV-2 variants were normalized by qRT-PCR and then infected cells for comparison. In this method, the normalization was focused on the "same amount of pseudovirus particles". Thus, the number of S proteins expressed on the surface of the pseudotyped virus may not be the same among these pseudoviruses, which may affect the entry capacities analysis. It is the limitation of this method[58].

Altogether, we elucidated the molecular mechanism of BA.2, BA.2.12.1, BA.4/5 S and BA.2.75 RBD in complex with hACE2 and BA.2 S in complex with ghACE2, mACE2 and RatACE2. The interspecies receptor recognition properties of BA.2 and BA.4/5 to 28 ACE2 orthologs were further evaluated. Our data reveals the structural bases for receptor binding and interspecies receptor recognition of BA.2, BA.2.12.1, BA.2.75 and BA.4/5.

## Methods

### Cells
HEK293F suspension-cultured cells (Gibco, 11625-019) were cultured at 37 °C in SMM 293-TII Expression Medium (Sino Biological, M293TII). HEK293T cells (ATCC, CRL-3216), Vero cells (ATCC, CCL-81) and BHK-21 adherent cells (ATCC, CCL-10) were cultured at 37 °C in Dulbecco's modified Eagle medium (DMEM) supplemented with 10% fetal bovine serum (FBS).

### Gene cloning
Gene encoding the ectodomain of the Omicron S protein (residues 14–1208) with 6 P mutants (F817P, A892P, A899P, A942P, K986P, and V987P) was fused with a C-terminal T4 fibritin trimerization motif, a Strep-tag II, and an 8×His tag and cloned into a mammalian cell expression vector pCAGGS. A Kozak sequence and an exogenous signal peptide derived from µ-phosphatase (MGILPSPGM-PALLSLVSLLSVLLMGCVAETGT) were added into the N terminus to maximize the protein production as previously reported[59].

The RBD (residues 319–541) of SARS-CoV-2 PT was fused with itself signal peptide in the N-terminus. The RBD (residues 319–541) of

Omicron sub-variants (BA.1, BA.2, BA.2.12.1, BA.2.75 and BA.4/5), BA.4/5-RBD-based mutants (BA.4/5-R452L RBD, BA.4/5-V486F RBD and BA.4/5-Q493R RBD) and BA.2-RBD-based mutant (BA.2-R493Q RBD) was fused with IL-10 signal peptide in their N-terminus, respectively. All of these RBDs harbored a Hexa-His tag sequence at the C-terminus. hACE2 (residues 1–615), mACE2 (residues 1–615), RatACE2 (residues 1–615), or ghACE2 (residues 1–615 was fused with a Hexa-His tag sequence at the C-terminus. All constructs were inserted into the pCAGGS vector for protein expression, respectively.

## Protein expression and purification

The separate pCAGGS plasmid encoding S, ACE2 or RBD protein was transfected using polyethyleneimine (PEI) and expressed in HEK293F cells. Cell culture supernatants were collected after a 4-day transfection. The supernatants containing the S proteins were purified using His-Trap HP columns (Cytiva) and the Superose™ 6 Increase 10/300 GL column (Cytiva). The RBDs and ACE2s were purified using His-Trap HP columns (Cytiva), followed by the Superdex™ 200 Increase 10/300 GL column (Cytiva). Purified proteins were stored in a buffer containing 20 mM Tris–HCl (pH 8.0) and 150 mM NaCl.

The S proteins produced by HEK293F cells were used for cryo-EM structures, and the RBDs were used for SPR and flow cytometry assay. For BA.2.75 RBD, it was also used for crystallization. The hACE2 protein produced by HEK293F cells was used for structural determination by cryo-EM or X-ray diffractions and binding affinity assay using SPR. The proteins for SPR assay were stored in PBST buffer (1.8 mM $KH_2PO_4$, 10 mM $Na_2HPO_4$ (pH 7.4), 137 mM NaCl, 2.7 mM KCl, and 0.005% (v/v) Tween 20).

## SPR assay

To determine the binding affinities of Omicron sub-variants towards hACE2, we conducted SPR experiments using a Biacore 8 K (Cytiva) at 25 °C in PBST buffer. The RBDs of SARS-CoV-2 (PT, BA.2, BA.2.12.1, BA.2.75, and BA.4/5) were diluted using the immobilization buffer and then immobilized on individual channels of a CM5 sensor chip using the Amine Coupling Kit (Cytiva, BR100633). Serially diluted hACE2 samples were flowed through the chip in single-cycle mode. Binding affinities represented by $K_D$ values were obtained with Biacore Insight Evaluation software v.3.0 (Cytiva) using the 1:1 binding model. The values indicate the mean ± standard deviations (SD) of three independent experiments.

To determine the binding affinities between ACE2 orthologs (mouse, rat, golden hamster, dog, cat, rabbit, sheep, horse, pig, and goat) and RBDs (BA.1, BA.2, BA.4/5, and a BA.2 mutant harboring R493Q reverse mutation), the ACE2s were diluted in the immobilization buffer and then immobilized on individual channels of a CM5 sensor chip using the Amine Coupling Kit (Cytiva, BR100633). Each RBD was serially diluted into five concentration gradients, which were flowed through the chip in single-cycle mode. Binding affinities represented by $K_D$ values were obtained with Biacore 8 K Evaluation Software (Cytiva) using the 1:1 binding model. The values indicate the mean ± SD of three independent experiments.

## Production and quantification of pseudoviruses

The protocols of a replication-deficient vesicular stomatitis virus (VSV) vector backbone (VSV-ΔG-GFP) was used to obtain pseudoviruses of SARS-CoV-2 sub-variants[58]. The plasmids containing genes of representative full-length spike proteins were transfected into HEK293T cells. The VSV-ΔG-GFP pseudoviruses were added to the cell plates 24 h later, and then the inoculum was removed after incubation for 1 h at 37 °C. After being washed with PBS, the cells were cultured in DMEM containing 10% FBS and anti-VSV-G mouse monoclonal antibody (10 μg/mL) produced by I1-Hybridoma (ATCC, CRL-2700). The pseudoviruses were harvested 30 h post-infection, filtered by 0.45 μm filters (Millipore, SLHP033RB), and stored at −80 °C.

Unpackaged RNA was removed by 0.5 U/μL BaseMuncher endonuclease (Abcam) at 37 °C for 1 h. The viral RNA was then extracted using an RNA extraction kit (Bioer Technology) and quantified by quantitative RT–PCR (qRT-PCR) using a 7500 fast real-time PCR system (Applied Biosystems). The L gene of VSV was quantified by primers and the probe, as previously described[60].

## Pseudovirus entry assay

The normalized pseudovirus particles for SARS-CoV-2 PT and variants were diluted to an equal amount. Then, each pseudovirus was added to a 96-well plate containing Vero cells (100 μL per well). After 15 h, the fluorescent cells representing the infection efficiency were counted using a CQ1 confocal image cytometer (Yokogawa). Each group contained 6 replicates, and all the analysis was repeated twice. The statistics were presented via GraphPad Prism 8.

## Flow cytometry assay

The plasmids encoding 29 full-length ACE2 orthologs (human, monkey, grivet, chimpanzee, gorilla, rabbit, mouse, rat, guinea pig, golden hamster, Malayan pangolin, cat, dog, horse, pig, fox, civet, mink, goat, sheep, camel, alpaca, bovine, little brown bat, fulvous fruit bat, greater horseshoe bat, Chinese horseshoe bat, least horseshoe bat and lesser hedgehog tenrec) fused with eGFP at the C-terminal were transfected into BHK-21 cells using PEI at the mass ratio of 1:3. Solutions containing the RBD (5 μg/mL) of BA.1, BA.2 or BA.4/5 were incubated with 29 ACE2 orthologs (including hACE2)-expressing BHK-21 cells at 37 °C for 1 h, respectively. Subsequently, cells were washed with PBS thrice and stained with anti-His tag mouse monoclonal antibody conjugated to APC (diluted at a ratio of 1:500, Miltenyi Biotec, 130-119-820) for 1 h before being analyzed using BD FACS CantoII Flow Cytometer (BD Biosciences). ACE2-transfected BHK-21 cells incubated with the N-terminal domain (NTD) of SARS-CoV-2 PT and BHK-21 cells transfected using CD26 fused with eGFP were used as negative controls. The data were analyzed using FlowJo v10.8 Software (BD Life Sciences). All experiments were performed at least three times; one representative of each experiment is shown in the Figure.

## Cyro-EM sample preparation and data acquisition

For the ACE2-bound S protein complex of BA.2, BA.2.12.1 or BA.4/5, the S protein was incubated with the purified hACE2 at a 1:4 molar ratio (S trimer to hACE2) overnight on ice, followed by purification by concentration and dialysis using an Ultracon concentrator (Millipore) with 20 mM Tris (pH 8.0) and 150 mM NaCl. A droplet (3.0 μL) of the purified complex at a concentration of 2.0–3.0 mg/mL was applied to glow-discharged C-flat R1.2/1.3 (300 mesh) holey carbon grids and subsequently vitrified using Vitrobot Mark IV (Thermo Fisher Scientific). For Omicron BA.2 S/hACE2, BA.2.12.1 S/hACE2, BA.4/5 S/hACE2, BA.2 S/mACE2, BA.2 S/RatACE2 and BA.2 S/ghACE2 complex datasets, 10317, 6768, 6282, 6628, 7164, and 8408 movies were collected respectively on a 300 kV Titan Krios transmission electron microscope equipped with a Gatan K3 detector and GIF Quantum energy filter. EPU software (Thermo Fisher Scientific) was used for automatic data collection. For BA.2 S/hACE2 and BA.2.12.1 S/hACE2 complexes, movies were collected with a calibrated pixel size of 0.85 Å. Cryo-EM data of BA.4/5 S/hACE2 complex was collected with a calibrated pixel size of 0.88 Å. For BA.2 S/mACE2 complex, BA.2 S/RatACE2 complex and BA.2 S/ghACE2 complex datasets, movies were collected with a calibrated pixel size of 0.67 Å, 0.88 Å and 0.88 Å, respectively. The defocus range was between −1.0 μm and −2.0 μm. Each movie was dose-fractionated into 32 frames with a total dose of 50 e⁻/Å².

## Image processing and 3D reconstruction for cryo-EM data

Super-resolution movies were corrected for drift using MotionCor2 v.1.4.2[61], and contrast transfer function (CTF) parameters were determined using CTF estimation in the patch mode. Particle picking and

extraction, 2D classification, Ab-initio reconstruction, heterogeneous refinement, CTF refinement, homogenous/non-uniform refinement, and local refinement were then performed using cryoSPARC v3.3.1[62]. The map was finally sharpened by DeepEMhancer v0.14[63].

For Omicron BA.2 S/hACE2 complex dataset, a total of 244,523 particles were extracted from the 10,317 micrographs after performing 2D classification, and then were used for initial reconstruction and heterogeneous refinement. A total of 141,550 particles were used for the iterative cycles of global CTF refinement, and a density map at 3.09 Å resolution was obtained. The RBD/hACE2 region was further performed local refinement and resulted to a density map at 3.14 Å resolution.

For Omicron BA.2.12.1 S/hACE2 complex dataset, a total of 150,981 particles were extracted from the 6768 micrographs after performing 2D classification, and then were used for initial reconstruction and heterogeneous refinement. 111,387 particles were used for the iterative cycles of global CTF refinement, and a density map at 3.19 Å resolution was obtained. The RBD/hACE2 region was further performed local refinement and resulted to a density map at 3.09 Å resolution.

For Omicron BA.4/5 S/hACE2 complex dataset, a total of 460,236 particles were extracted from the 6,282 micrographs after performing 2D classification, and then were used for initial reconstruction and heterogeneous refinement. 355,600 particles were used for the iterative cycles of global and local CTF refinement, and a density map at 2.58 Å resolution was obtained. The RBD/hACE2 region was further performed local refinement and resulted to a density map at 2.66 Å resolution.

For Omicron BA.2 S/mACE2 complex dataset, a total of 168,410 particles were extracted from the 6,628 micrographs after performing 2D classification, and then were used for initial reconstruction and heterogeneous refinement. A total of 91,073 particles were used for global and local CTF refinement, and a density map at 3.02 Å resolution was obtained after performing non-uniform refinement. The RBD/mACE2 region was further performed local refinement and resulted to a density map at 3.20 Å resolution.

For Omicron BA.2 S/RatACE2 complex dataset, a total of 658,163 particles were extracted from the 7164 micrographs after performing 2D classification, and then were used for initial reconstruction and heterogeneous refinement. A total of 475,227 particles were then imported to RELION-3.1[64] for RBD-focused 3D classification, and 50,491 particles were further performed homogenous refinement in cryoSPARC v3.3.1[62], and a density map at 3.01 Å resolution was obtained. The RBD/RatACE2 region was then performed local refinement and resulted to a density map at 3.29 Å resolution.

For Omicron BA.2 S/ghACE2 complex dataset, a total of 297,170 particles were extracted from the 8,408 micrographs after performing 2D classification, and then were used for initial reconstruction and heterogeneous refinement. 224,458 particles were used for global and local CTF refinement, and a density map at 2.96 Å resolution was obtained after performing non-uniform refinement. The RBD/ghACE2 region was further performed local refinement and resulted to a density map at 2.94 Å resolution.

## Model building
The structure of the RBD/hACE2 region in the S/hACE2 complex (PDB: 7KNB) was docked into the cryo-EM density maps obtained by local refinement for the Omicron BA.2 S/hACE2, BA.2.12.1 S/hACE2, BA.4/5 S/hACE2, BA.2 S/mACE2, BA.2 S/RatACE2 and BA.2 S/ghACE2 complexes using UCSF Chimera v.1.14[65], respectively. The models were manually corrected and refined iteratively using Coot v.0.9.8 and Phenix v.1.19.2[66,67]. The stereochemical quality of each model was evaluated using MolProbity[68].

Data collection, processing, and refinement statistics are summarized in Supplementary Table 2. All structural Figures were generated using ChimeraX v.1.3[69] and PyMOL v.2.4 (https://pymol.org/2/) softwares.

## Crystallization, data collection and structure determination for the BA.2.75 RBD/hACE2 complex
Crystallization trials were performed using the sitting drop vapor-diffusion method with commercial crystallization kits (Hampton Research and Molecular Dimensions). 1 µL purified BA.2.75 RBD/hACE2 complex at the concentration of 5 mg/ml or 10 mg/ml was mixed with 1 µL reservoir solution. The resultant drop was then sealed, equilibrating against 100 µL reservoir solution at 4 or 18 °C. Diffractable crystals were obtained in 0.15 M Ammonium sulfate, 0.1 M Sodium HEPES (pH 7.0), 20% w/v PEG 4000 at 18 °C.

Crystals were flash-cooled in liquid nitrogen after a brief soak in reservoir solution with the addition of 20% (v/v) glycerol. X-ray diffraction data were collected under cryogenic conditions (100 K) at Shanghai Synchrotron Radiation Facility (SSRF). The BA.2.75 RBD/hACE2 dataset was collected at BL02U1. The data were indexed, integrated, and scaled with HKL2000[70]. The structure was then determined by the molecular replacement method using Phaser[71] with a SARS-CoV-2 RBD/hACE2 molecule (PDB: 6LZG), followed by refinement using Coot v.0.9.8[67] and phenix.refine in Phenix v.1.19.2[66]. The stereochemical qualities of the final models were assessed with MolProbity[68]. Data collection, processing, and refinement statistics are summarized in Supplementary Table 3. All structural Figures were generated using ChimeraX v.1.3[69] and PyMOL v.2.4 (https://pymol.org/2/) softwares.

## Reporting summary
Further information on research design is available in the Nature Portfolio Reporting Summary linked to this article.

## Data availability
The data that support this study are available from the corresponding authors upon request. The cryo-EM maps have been deposited in the Electron Microscopy Data Bank (EMDB) under accession codes EMD-33870 (Cryo-EM structure of SARS-CoV-2 Omicron BA.2 RBD in complex with human ACE2 (local refinement)), EMD-33841 (Cryo-EM structure of SARS-CoV-2 Omicron BA.2.12.1 RBD in complex with human ACE2 (local refinement)), EMD-34409 (Cryo-EM structure of SARS-CoV-2 Omicron BA.4/5 RBD in complex with human ACE2 (local refinement)), EMD-34138 (Cryo-EM structure of SARS-CoV-2 Omicron BA.2 RBD in complex with mouse ACE2 (local refinement)), EMD-34217 (Cryo-EM structure of SARS-CoV-2 Omicron BA.2 RBD in complex with rat ACE2 (local refinement)), EMD-34120 (Cryo-EM structure of SARS-CoV-2 Omicron BA.2 RBD in complex with golden hamster ACE2 (local refinement)), EMD-34494 (Cryo-EM map of SARS-CoV-2 Omicron BA.2 spike trimer in complex with human ACE2 (three-RBD-up conformation)), EMD-34498 (Cryo-EM map of SARS-CoV-2 Omicron BA.2.12.1 spike trimer in complex with human ACE2 (three-RBD-up conformation)), EMD-34499 (Cryo-EM map of SARS-CoV-2 Omicron BA.2 spike protein in complex with mouse ACE2), EMD-34509 (Cryo-EM map of SARS-CoV-2 Omicron BA.4/5 (N658S) spike trimer in complex with human ACE2 (three-RBD-up conformation)), EMD-34510 (Cryo-EM map of SARS-CoV-2 Omicron BA.2 spike trimer in complex with golden hamster ACE2) and EMD-34506 (Cryo-EM map of SARS-CoV-2 Omicron BA.2 spike trimer in complex with rat ACE2). The atomic structure coordinates were deposited in the RCSB Protein Data Bank (PDB) under the accession codes 7YJ3 (Cryo-EM structure of SARS-CoV-2 Omicron BA.2 RBD in complex with human ACE2 (local refinement)), 7YHW (Cryo-EM structure of SARS-CoV-2 Omicron BA.2.12.1 RBD in complex with human ACE2 (local refinement)), 8H06 (Cryo-EM structure of SARS-CoV-2 Omicron BA.4/5 RBD in complex with human ACE2 (local refinement)), 7YVU (Cryo-EM structure of SARS-CoV-2 Omicron BA.2 RBD in complex with mouse ACE2 (local refinement)), 8GRY (Cryo-EM structure of SARS-CoV-2 Omicron BA.2 RBD in complex with rat ACE2 (local refinement)), 7YV8 (Cryo-EM structure of SARS-CoV-2 Omicron BA.2 RBD in complex with golden

hamster ACE2 (local refinement)) and 8H5C (Structure of SARS-CoV-2 Omicron BA.2.75 RBD in complex with human ACE2). Additional data has been deposited in China National Microbiology Data Center (NMDC) with accession numbers NMDCS0000020 (Cryo-EM structure of SARS-CoV-2 Omicron BA.2 RBD in complex with human ACE2 (local refinement)), NMDCS0000021 (Cryo-EM structure of SARS-CoV-2 Omicron BA.2.12.1 RBD in complex with human ACE2 (local refinement)), NMDCS0000025 (Cryo-EM structure of SARS-CoV-2 Omicron BA.4/5 RBD in complex with human ACE2 (local refinement)), NMDCS0000023 (Cryo-EM structure of SARS-CoV-2 Omicron BA.2 RBD in complex with mouse ACE2 (local refinement)), NMDCS0000024 (Cryo-EM structure of SARS-CoV-2 Omicron BA.2 RBD in complex with rat ACE2 (local refinement)), NMDCS0000022 (Cryo-EM structure of SARS-CoV-2 Omicron BA.2 RBD in complex with golden hamster ACE2 (local refinement)) and NMDCS0000026 (Structure of SARS-CoV-2 Omicron BA.2.75 RBD in complex with human ACE2). Other structures for analysis, including 7KNB [https://doi.org/10.2210/pdb7knb/pdb] and 6LZG [https://doi.org/10.2210/pdb6lzg/pdb], were obtained from the PDB. Source data are provided with this paper.

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

## Acknowledgements

We are grateful to the PMI-IMCAS Public Technology Service Center for its support on the FACS assay. We thank Y.Q.M. and X.X.B. at the Cryo-EM Center, Shanxi Academy of Advanced Research and Innovation for their technical support on the Cryo-EM. We thank the staff of the BL02U1 beamline at Shanghai Synchrotron Radiation Facility. This work was supported by the National Key R&D Program of China (2023YFC3041500 and 2021YFC2301401 to J.Q.), Strategic Priority Research Program of the Chinese Academy of Sciences (XD29010202 to G.F.G.), the National Natural Science Foundation of China (92169208 to J.Q.), the Special Program of China National Tobacco Corporation (110202102034 to J.Q.), the fellowship of China Postdoctoral Science Foundation (2022T150688 to K.L.) and Young Elite Scientists Sponsorship Program by CAST (2021QNRC001 to K.L.).

## Author contributions

J.Q. and G.F.G. initiated and designed the project. B.B., J.Z. and W.L. purified the proteins with the help of Y.M., X.X.L., X.W. and Z.X. B.B., C.L. and L.L. performed the SPR analysis with the help of W.Z. and Z.F. L.L. conducted the flow cytometry assay. Q.J. collected the X-ray data with the help of O.L. Z.Z. and Y.X. performed cryo-EM sample preparation, data collection, image processing, and map reconstruction with the assistance of X.M.L. and J.S. J.Q. and Y.C. conducted model building and refinement. D.L. and X.Z. performed pseudovirus-related assays. L.W. and J.M. analyzed the prevalence of Omicron sub-variants. K.L., Z.Z., Y.S. and X.Z. analyzed the data. Z.Z., G.S., K.L., J.Q, P.W. and G.F.G. wrote and revised the manuscript.

## Competing interests

The authors declare no competing interests.
