## [Peer Review File · Nature Communications]

Structural basis for receptor binding and broader interspecies receptor recognition of currently circulating Omicron sub-variantsReviewers' Comments:

Reviewer #1:

Remarks to the Author:

The manuscript "Structural basis for receptor binding and broader interspecies receptor recognition of currently circulating Omicron sub-variants" by Zhao and colleagues investigated the binding capacities of human and 28 animal ACE2 orthologs to RBDs of SARS-CoV-2 Omicron and its sub-variants, and also performed binding analysis for specific single mutations, such as the amino acid at position 493 as Arg or Gln, and revealed that R493Q reverse mutation enhanced the bindings to ACE2s from humans and animals in close environment of human life which is important for evaluation of increased risk of cross-species transmission. The authors carried out an impressive amount structural analysis by cryo-EM or x-ray crystallography for Omicron variant spike or RBD in complex with ACE2 of human, mouse, rat or golden hamster to reveal the structural basis for receptor binding and interspecies recognition. Built on their previous study of Omicron BA.1 recognition to ACE2s from multiple species, this manuscript expanded to provide more detailed interspecies recognition analysis on the recent emergent Omicron sub-variants, and is important for the monitoring of interspecies transmission of virus with pandemic potential.

This manuscript is well written, I only have some minor comments on data presentation:

1. Figure 3F, it might be great to show the EM density for the alternative conformation of His34 in a supplementary figure.
2. For the heatmap in Figure 5, is it possible to use the same orientation as in Panel A.
3. For KD and related SD, and the expression of numbers in Table S5 and S6, it should be unified to one format, eg, nM and E-notation (such as 2.00E+6). Current forms apparently came from two authors.
4. The full name/abbreviation "PT" should be moved to line 109.

Reviewer #2:

Remarks to the Author:

Zhao and collaborators report in this work the binding affinities of human ACE-2 with RBD of SARS-CoV-2 Omicron variants. They also compare the binding of these RBD to ACE-2 from several animals. The emergence of the Omicron variant has marked a new pace in the evolution of the SARS-CoV-2 pandemic. This variant, that spread rapidly and has evolved in multiple sub-variants, has accumulated an extremely high number of mutations compared to the previous circulating VOCs, suggesting an evolution as escape mutant (loss of binding by several neutralizing antibodies) but also a potential for interspecies transmission.

Both aspects have been largely studied in the literature and the structure of several Omicron sub-variant reported in this paper have been already published (Xu et al. Cell Research 2022; Cao et al. Nature 2022). The binding specificity toward ACE-2 orthologs for some of the variants reported in this paper has been reported by others (Xu et al. 2022) and the by the authors of this work too (Li et al. Cell Discovery 2022). Interestingly the results reported in this paper are not always in agreement with what has been previously published. This point deserves more critical analysis in the discussion. While the comparative analysis of the interaction of the Omicron variants with different ACE-2 is a valuable study the results presented in this work are merely listed and would need to be interpreted in the context of their potential impact on the evolution of the pandemic.

More in detail:

1) Lines 134-139. To compare the "entry capabilities" of the different variants using pseudotyped virus it is necessary to normalize on the total amount of spike. The spike of different variants can be differently incorporated in the pseudoviruses so normalization based only on the infectivity of D614G is not sufficient. If we do not know how much spike is present on the particles for the different variants, we cannot correlate with their ability to mediate entry.

- 2) In the supplementary tables use the same order of magnitude for the values, matching the numbers presented in the figures. Also, would it be possible to represent the BLI experiments as aligned data in x and y to facilitate the comparison between the different samples, as for example presented in Xu et al. 2022?
- 3) Line 169. Suppl. Table 4 is dense. May be color coding the residues that are not present in all the variants may facilitate visualization (i.e. first row, S19 highlight A475 but not N477).
- 4) Lines 182-184. Clarify the role of charges in the binding affinity. As reported by the authors in a previous paper (Han et al. cell 2022) Omicron mutations increase the positive charges of the region interacting with the negative charged area of ACE-2. Here the author found that the R493Q reverse mutation weaken the positive charges in two variants but these variants (BA.2.75 and BA.4/5 - Q493) show higher affinity than variant BA.2 (R493).
- 5) Lines 301-308. The authors need to comment on the different results reported in this paper compared to published results. If the differences between X-ray analysis and cryo-EM may be explained by the "limitation of crystallography" the differences by cryo-EM analysis need to be critically analyzed.
- 6) Line 313. It is not clear what the authors mean with "resembling the fact..." may be "similarly to" would fit better.
- 7) Lines 327-334. Discuss and clarify the differences with the other studies and the role the binding affinity may play in interspecies transmission.
- 8) Line 373. Transfection not infection.
- 9) Some of the data have been presented in previous publications by the same authors but with sometimes different results. For example:
Fig.4 FACS analysis: The plots for BA.1 look different in this work and in Li et al. (Cell Discovery 2022). For mink and pig no binding is reported in Li et al. whereas the data presented here show binding, similarly there is a slight difference for the little brown bat and fulvous fruit bat data.

Point-by-point letter

REVIEWER COMMENTS

Reviewer #1 (Remarks to the Author):

The authors carried out an impressive amount structural analysis by cryo-EM or x-ray crystallography for Omicron variant spike or RBD in complex with ACE2 of human, mouse, rat or golden hamster to reveal the structural basis for receptor binding and interspecies recognition. Built on their previous study of Omicron BA.1 recognition to ACE2s from multiple species, this manuscript expanded to provide more detailed interspecies recognition analysis on the recent emergent Omicron sub-variants, and is important for the monitoring of interspecies transmission of virus with pandemic potential.

Response: Thanks for your positive comments.

This manuscript is well written, I only have some minor comments on data presentation:

1. Figure 3F, it might be great to show the EM density for the alternative conformation of His34 in a supplementary figure.

Response: Thanks for the suggestions. We have shown the EM density for the alternative conformation of His34 in the supplementary Figure 5d.

2. For the heatmap in Figure 5, is it possible to use the same orientation as in Panel A.

Response: Thanks for the suggestions. The heatmap has the same orientation as in Panel A in Figure 5 of the revised version.

3. For K_D and related SD , and the expression of numbers in Table S5 and S6, it should be unified to one format, eg, nM and E-notation (such as 2.00E+6). Current forms apparently came from two authors.

Response: Thanks for your reminding. The revised version has unified the expression of numbers for K_D and SD .

4. The full name/abbreviation “PT” should be moved to line 109.

Response: Given that the full name/abbreviation of “PT” was introduced in the introduction section (line 60), “PT” was used in the results section (line 110).

Reviewer #2 (Remarks to the Author):

....

Both aspects have been largely studied in the literature and the structure of several Omicron sub-variant reported in this paper have been already published (Xu et al. Cell Research 2022; Cao et al. Nature 2022). The binding specificity toward ACE-2 orthologs for some of the variants reported in this paper has been reported by others (Xu et al. 2022) and the by the authors of this work too (Li et al. Cell Discovery 2022). Interestingly the results reported in this paper are not always in agreement with what has been previously published. This point deserves more critical analysis in the discussion. While the comparative analysis of the interaction of the Omicron variants with different ACE-2 is a valuable study the results presented in this work are merely listed and would need to be interpreted in the context of their potential impact on the evolution of the pandemic.

Response: Thanks for the comments. Xu et al. reported the BA.2 spike/hACE2, BA.1 spike/mouse ACE2, and BA.2 spike/mouse ACE2 complex structures and the binding specificity towards several ACE2 orthologs (human, mouse, cat, rat, and dog) for PT, BA.1 and BA.2 variants, and other three ACE2 orthologs (horse, sheep and pig) for PT and BA.1. Cao et al. determined the structures of apo spikes from BA.1, BA.2, BA.3, BA.2.13, BA.2.12.1 and BA.4/5 but not the structures of these spikes complexed with hACE2. Our previous work (Li et al. Cell Discovery 2022) evaluated the interspecies recognition of the Omicron BA.1 and Delta RBDs by 27 ACE2 orthologs, including humans. In this paper, we reported seven complex structures that are BA.2 spike/hACE2, BA.2.12.1 spike/hACE2, BA.2.75 RBD/hACE2, BA.4/5 spike/hACE2, BA.2 spike/mouse ACE2, BA.2 spike/rat ACE2 and BA.2 spike/golden hamster ACE2, and measured the binding specificity towards 29 species for BA.1, BA.2 and BA.4/5 using FACS and SPR assays. This is the continuous efforts from our lab based on our previous work and a comprehensive comparative study, including the work by Xu et al. and Cao et al. Following this reviewer's suggestions, we have extensively discussed all the relevant literature (lines 301-329). The potential impact of these Omicron variants on the evolution of the pandemic was also discussed (lines 330-344).

More in detail:

1) Lines 134-139. To compare the “entry capabilities” of the different variants using pseudotyped virus it is necessary to normalize on the total amount of spike. The spike of different variants can be differently incorporated in the pseudoviruses so normalization based only on the infectivity of D614G is not sufficient. If we do not know how much spike is present on the particles for the different variants, we cannot correlate with their ability to mediate entry.

Response: Thanks for the comments. We used the VSV-based pseudovirus system based on the previous protocols (Nie et al. Nature Protocols 2020). The pseudoviruses harboring S protein from different SARS-CoV-2 variants were normalized by qRT-PCR and then infected cells for comparison. In this method, the normalization was focused on the “same amount of pseudovirus particles”. So, as pointed out by the reviewer, the number of S proteins expressed on the surface of the pseudotyped virus may not be the same among these pseudoviruses, which may affect the entry capacities analysis. It is the limitation of this method mentioned by Nie et al. We have added this limitation in the Discussion (Lines 367-373).

2) In the supplementary tables use the same order of magnitude for the values, matching the numbers presented in the figures. Also, would it be possible to represent the BLI experiments as aligned data in x and y to facilitate the comparison between the different samples, as for example presented in Xu et al. 2022?

Response: Thanks for the suggestions. Our study used a surface plasmon resonance (SPR) assay, not the BLI experiment, to test the binding affinities. $K_D=kd/ka$, which is related to the association rate constant (ka) and dissociation rate constant (kd) but not directly related to the response unit. To facilitate the comparison, we have listed the binding affinities in Figure 5b and presented the related heatmap in Figure 5c at the same orientation as in Figure 5a of the revised version.

The SPR assay in Figure 1d in Cao et al. Nature 2022 and BLI experiments in Figure 3a in Xu et al. Cell Research 2022 are also not presented as aligned data.

3) Line 169. Suppl. Table 4 is dense. May be color coding the residues that are not present in all the variants may facilitate visualization (i.e. first row, S19 highlight A475 but not N477).

Response: Thanks for the suggestions. Interacting residues in each row that are not shared

by all the variants are colored red in the revised version.

4) Lines 182-184. Clarify the role of charges in the binding affinity. As reported by the authors in a previous paper (Han et al. cell 2022) Omicron mutations increase the positive charges of the region interacting with the negative charged area of ACE-2. Here the author found that the R493Q reverse mutation weaken the positive charges in two variants but these variants (BA.2.75 and BA.4/5 – Q493) show higher affinity than variant BA.2 (R493).

Response: Thanks for the comments. The previous paper (Han et al. Cell 2022) described that **“R493 and R498 of RBD might be attracted by the negative charges around E35 and D38 of hACE2, respectively”**, which was based on structural analyses. The SPR assay in our another work showed that PT Q493R mutation decreases the binding affinity to hACE2 (Li et al. Cell Discovery 2022), which is consistent with this paper. In addition, the BA.2 RBD/hACE2 and BA.4/5 RBD/hACE2 complex structures in this paper further reveal the structural basis of enhanced binding affinity of R493Q substitution to hACE2; that is, R493Q reverse mutation makes K31 and its near residues of hACE2 more approach the RBD and form more vdw contacts and H-bonds (Supplementary Table 4). This part was discussed in lines 319-329.

5) Lines 301-308. The authors need to comment on the different results reported in this paper compared to published results. If the differences between X-ray analysis and cryo-EM may be explained by the “limitation of crystallography” the differences by cryo-EM analysis need to be critically analyzed.

Response: Thanks for the suggestions. We have commented on the results in this paper compared to those published (lines 301-329), in which the differences by cryo-EM analysis were analyzed (lines 305-314).

6) Line 313. It is not clear what the authors mean with “resembling the fact...” may be “similarly to” would fit better.

Response: Thanks for the suggestions. We have used “similarly to” to replace “resembling the fact...” in the revised version.

7) Lines 327-334. Discuss and clarify the differences with the other studies and the role the binding affinity may play in interspecies transmission.

Response: Thanks for the suggestions. The differences with the other studies have been discussed in lines 305-329. The roles the binding affinity may play in interspecies transmission have also been discussed in lines 330-344.

8) Line 373. Transfection not infection.

Response: Thanks for your reminding. It has been corrected.

9) Some of the data have been presented in previous publications by the same authors but with sometimes different results. For example:

Fig.4 FACS analysis: The plots for BA.1 look different in this work and in Li et al. (Cell Discovery 2022). For mink and pig no binding is reported in Li et al. whereas the data presented here show binding, similarly there is a slight difference for the little brown bat and fulvous fruit bat data.

Response: We appreciate your serious and rigorous comments. In our previous work (Li et al. Cell Discovery 2022), cells expressing pig ACE2 or mink ACE2 bind to BA.1 RBD with weakly positive but not negative. The FACS results in that paper (Li et al. Cell Discovery 2022) were described as “Briefly, monkey, rabbit, cat, fox, dog, raccoon dog, mink, pig, wild Bactrian camel, alpaca, bovine, goat, and sheep ACE2 orthologs were bound to all three RBDs tested using the flow cytometry assay”. In addition, we checked our original data and repeated these FACS experiments, which showed that pig ACE2 binds to BA.1 RBD weakly. Mink ACE2 also showed a weak binding to BA.1 RBD in the repeated experiments, which differs from our previously submitted data but is consistent with our previously reported data in Li et al. Cell Discovery 2022. For the little brown bat and fulvous fruit bat, there are some non-specific positive cells, as shown in the following Figure. These panels were replaced with new data in the revised version.

Reviewers' Comments:

Reviewer #1:

Remarks to the Author:

The authors have addressed all my comments in the revised manuscript.